# Fine Art Insurance Policies and Risk Perceptions: The Case of Malta

**Luke Pavia** [1], **Simon Grima** [1,*], **Inna Romanova** [2] **and Jonathan V. Spiteri** [1]

1   Department of Insurance, Faculty of Economics, Management, and Accountancy, University of Malta, MSD2080 Msida, Malta; luke.pavia.16@um.edu.mt (L.P.); jonathan.v.spiteri@um.edu.mt (J.V.S.)
2   Department of Finance and Accounting, Faculty of Business, Management and Economics, University of Latvia, LV-1586 Riga, Latvia; inna.romanova@lu.lv
*   Correspondence: simon.grima@um.edu.mt

**Abstract:** The aim of this paper is to identify the risks that need to be addressed when holding fine art, determine which are perceived as being the most important, and whether the risk perception is influenced by demographic variables such as age, educational background, and field of occupation. To identify the risks and evaluate the risk perception, we used a purposely designed questionnaire and sent it via various sources of communication systems and applications to individuals knowledgeable on fine arts. Findings revealed that, generally, art deterioration, art fraud, and art theft are the three main highlighted risks, with art deterioration considered in the high-risk range. In terms of risk perception, forgery is the biggest concern. On the other hand, considerations of the investment value of art lessened perceived risk exposure. Furthermore, the study has shown that certain risk perceptions were influenced by the participants' demographic variables. Both the identified risks and risk perception considerations analyzed within this study provide us with insights as to what needs to be considered when offering fine art insurance, particularly when it comes to which risks that are perceived as being the most pressing by potential policyholders, and how these perceptions vary according to individual demographics variables as noted above.

**Keywords:** fine art insurance; insurance policy; risk perception; risks

## 1. Introduction

The insurance industry has been growing over the past few years, with more and more people seeking to be protected from the unknown, especially in the case of highly valued items (Friedman et al. 2020). Rubin (2000) defines insurance as a "legally binding unilateral agreement between an insured and an insurance company to indemnify the buyer of a contract under specified circumstances. In exchange for premium payment(s), the company covers stipulated perils" (Rubin 2000). Without insurance, people would be exposed to relatively high losses in case any unforeseeable event that might happen.

Over the last two decades, there has been an exponential growth in the international art market (Adam 2014). In fact, art can in certain cases, be considered an 'investment'. In many countries there is a market that specifically caters for the insurance of such an asset, ranging from private collections to those in art galleries and museums (Fischer and Arnold 2010). However, in some other countries, such as Malta, no fine art insurance policies are offered.

Malta is a country with an accumulation of a varied richness of assets in fine arts. Given the country's art richness, particularly in public places with high public accessibility, such as churches and museums, it is unclear why such an area of potentially good revenue for insurance companies and brokers has not yet been exploited. The recent spate of thefts of works of art from private residences and churches (Xuereb 2019; Calleja 2006) highlights the need to research the topic of insuring fine arts against thefts extensively.

To the best of our knowledge, little has been written about the important factors and perceived risks to consider when offering and pricing fine art insurance policies.

Therefore, we aim to identify the risks that need to be addressed when holding fine art, determine which are perceived as being the most important and whether the risk perception is influenced by demographic variables. The study is based on the purposely designed questionnaire sent to individuals knowledgeable on fine arts. Two research questions are posed:

**RQ1.** *What are the risks that need to be addressed when holding art and which risks are perceived as being most prevalent?*

**RQ2.** *How do demographic variables affect people's perception of art risks and art insurance?*

In this paper, we focus on key demographics such as age, level of education, or field of study on the risk and valuation risk exposure ratings. We have chosen these particular variables since, as noted in Bezzina and Grima (2012) and Bezzina et al. (2014), they are the variables that generally have an effect on perceptions of risk and therefore on the demand which will have an effect on the potential entry of Insurers in this market and the value of the risk. More specifically, we analyze demographics such as age e.g., (Savage 1993), educational background e.g., (Sund et al. 2017), and academic discipline e.g., (Weisenfeld and Ott 2011). Knowing the demographics of a country and relating this to perceived risks of fine arts will enable insurers to determine the potential clientele, the value this clientele gives to the fine art, and in turn the premium to be charged for such a policy. This would help in determining the possible market demand for an insurance policy on fine art and therefore its pricing/perceived value.

Since as noted above, Malta is a country rich in both culture and art, it provides a noteworthy market potential for the fine art insurance and the study findings can be used by the Maltese insurance companies when offering and pricing fine art insurance policies to individuals. Moreover, similar to other studies of islands and small countries carried out by authors such as King (1993), Briguglio (1995), Magri et al. (2019), Xuereb et al. (2019), and Bezzina et al. (2014), Malta can be used as a contained study for insurance companies in larger regions and countries.

Currently, the only policy available in the Maltese market that could potentially and maybe indirectly cover artwork is the property insurance policy; art may be included under the contents section of the policy. However, this is very limited in scope and covers up to a limit, which is too low in value to cover artworks. Internationally, there is the same limitation with the contents section of the policy, however, in the foreign market, there are specific policies catering for fine art losses (MiniCo Insurance Agency LLC 2020).

## 2. Literature Review

A key feature that characterizes humankind is that of creation of and response to works of art. For the insurance industry, art can mean anything from paintings and sculptures, to instruments, clocks, and antiques (Reed 2017). Fine art usually refers to works that were created for the sole purpose of capturing the creative expression of the artist at the time of making it, as opposed to those pieces that are commissioned by someone for the purpose of craft or decorating. In the past it mainly related to paintings, sculpture, architecture, music, and poetry; mainly for the purpose of artists expressing themselves and creating something that was aesthetically pleasing. In today's art world, fine arts may also include film, photography, and printmaking (Delacare 2016).

### 2.1. Problems/Risks Faced by Fine Art and Objective of Having Insurance

As other assets, fine art is also exposed to a number of risks. Bialynicka-Birula (2013) categorizes the risks relating to the art into a number of categories, mainly including risks directly relating to the works of art, risks relating to the parties involved in the transaction of art, auctioneer or agent risks, market risks, and distant environment risks such as economic, force majeure, and crime risks (Bialynicka-Birula 2013).The main aspects identified through these risks can be highlighted as being art fraud, thefts, deterioration, and fluctuations in the value of the art, as well as 'legal issues arising from the owned

artworks' and 'changes in the operation of the art market' or 'the interpretation of art value', 'availability of information', and such changes (Thum-Thysen et al. 2017).

One of the main risks related to fine art is art 'fraud/forgery'. Elements of forgery or imitation art have been around for thousands of years, some dating way back to Ancient Rome. The main drive for forgeries, apart from the obvious monetary incentive, is the fact that in most cases, the forger's own original art was not appreciated or recognized, and this would lead them to duping the art world as a way of seeking revenge (Ricci 2019). Another substantial fine art risk is 'art thefts'. A number of famous art pieces have been stolen throughout history, including The Scream (Edvard Munch 1893) and Mona Lisa (Michelangelo Da Vinci, 1503–1506) to mention but a few. In Malta the most famous art theft relates to Caravaggio's St Jerome Writing from St John Co-Cathedral in Valletta. The work was eventually recovered after the negotiations between the thieves and the authorities led to the thieves and the place where the work was being kept (Camilleri 2015).

'Damage and deterioration' are seen as one of the risks faced by the fine art. A narrative told by John Pope-Hennessy recounts the tale of the shocking flooding which damaged a number of art pieces in Florence in 1966, which caused widespread destruction. Thousands of works of art were damaged and needed some extent of restoration. Even to this day, many pieces still require conservation and restoration works, however, luckily, not many artworks were completely damaged beyond repair (Anon 2004; McNearney 2019). One of the later cases of damaged art is to the famous cathedral, Notre-Dame de Paris, France, whereby in April of 2019, the restoration works being completed on the structure caused a fire to break out. The over 800-year-old cathedral survived many potentially detrimental events such as the French Revolution and the Second World War, only to be crumbled by a fire (Fire at Cathédrale Notre-Dame de Paris) (Barrau 2019) caused by the equipment which was meant to improve and better the structure (Ferreira 2019).

'Loss of art' is another problem in the fine art market. When art is lost, this could be due to thefts, damage, or even simply misplacing of the piece. There is also experimental art which uses materials which are not conventional or tried and tested. This type of art would thus only live in the moment, and can never truly be seen again, except in photographs, videos, or mementos from the event (Charney 2018). Such cases are the subject of discussion in relation to whether these may be insured due to the inevitable consequence of the experimental use of material.

Another risk is related to problems and court cases due to the legal issues arising from the ownership of the art risk. Cases such as those of Menzel v. List (1969) and O'Keeffe v. Snyder (1980) both ruled in favor of the plaintiff, even though the abandonment of the pieces due to war risk (for Menzel) and the thefts (for O'Keeffe) had happened over 25 and over 30 years ago, respectively. Both these cases were tried in New York, where the principle is usually to return goods to the original owner, no matter the situation (Danziger and Danziger 2010). Laws such as those in Sweden, however, decree that claims from the original owners or heirs must be made within 5 years of its sale. It would, therefore, benefit plaintiffs to claim under laws like that of New York, while defendants to bring the case forward in law jurisdictions like that of Sweden (Yip 2010).

Changes in the Art Market/Fluctuations in Value of Art are related to another important risk. In the early 2010s, there was a significant percentage increase in auction sales from a decade prior (Adam 2014; Petterson 2014). These transactions only cover those sales made during auctions, and not those made by dealers or online transactions. The auction price may not even be the true value of the piece, merely the subjective value of the individual who bids on it. Even the type of art that people are going for has shifted and transformed. This change comes from a number of factors, including availability of the artworks, change in the collector wants or needs, the idea of prestige that comes from buying art, and the purchasing of art as a means of investment. As newer and wealthier players enter the market, the already dynamic market progresses and adapts, causing further changes in the market (Petterson 2014).

*2.2. What Insurances Are Available for Fine Art?*

People view insurance in many ways; some people view it as a definite price being paid for a possible large loss to be covered which highlights the transferring of one's risk, while others see it as having one's risk reduced by pooling funds of the many to cater for the claims of the few, and by doing so, sharing the risk (Vaughan and Vaughan 2013). The Insurance Business Act, Chapter 403 of the Laws of Malta, defines an insurance contract to be an agreement where an insurer decides to take on a risk at a premium, and in turn the insurer pays the insured's loss, damage or liability for which an element of uncertainty is present (Insurance Business Act Cap 1998, Chapter 403). Insurance has in fact been in existence for a couple thousand years, a form of it being first prevalent in the spreading of marine risk. However, non-life insurance as we know it today started out in Genoa, an Italian city with several marine-based trading happening at the time and the notion of risk spreading was then adopted by other parts of Europe and ultimately the New World (Lester 2009). Non-life insurance contracts, which insure physical and monetary risks, are typically for a period of up to a year and renewed based on any updated information regarding the risk (Lester 2009). People have turned to art as a form of investment and as a way to protect their assets. 'Art, as a means of investment', works differently to the way stocks do; there are no dividends involved and no true book value. For this reason, investing in art is not for everyone (Karabell 2011). Fine art insurance has been a niche market in the industry for a long time (Gristina 2014).

The British Antique Dealers' Association emphasize the need for security in order to protect the artworks an individual owns. They even make recommendations on the literature from certain security companies that members should use, which include what locks and alarms can be used. Some crime prevention specialist and art insurance professionals may also provide advice in this regard. As a backup to the security that the individual has in place, they may also choose to take out insurance cover, not as a substitute but as further reassurance in case the security measures were to fail. While some contents can be included in a standard property insurance policy, for individuals with a substantial value of artworks, they are likely to be better served by a specific policy which will provide more extensive cover. This type of cover will typically be provided by insurance brokers which will then liaise with the insurer themselves (The British Antique Dealers' Association 2009).

Art insurance can be described as the policy taken out to cover for all risks associated with the artworks under the ownership of the policyholder. This includes accidental damage, theft, and other natural causes; and also has certain exclusions and conditions (Reed 2017). With the ever-growing and evolving art market, the fine arts insurance market has matured and advanced too. A list has been created by certain companies to define what is captured under works of art, which includes: paintings, ceramics, drawings, furniture, photographs, books, manuscripts, sculptures, and many more. Considering the fact that valuing a piece may prove to be very difficult, specialists are assigned this task (Rodríguez 2009).

## 3. Research Design and Methods

*3.1. Building the Questionnaire*

To carry out the study we used a mixed-method case study approach on Malta and the Maltese people, adapting a framework suggested by Yin (2002) and Stake (1995). We identified propositions on risks related to fine art, from literature and from preliminary discussions with specialists in the field by applying a thematic analysis as suggested by Braun and Clarke (2006). The choice of the demographic variables wass based on Bezzina and Grima (2012) and Bezzina et al. (2014), Savage (1993), Sund et al. (2017), Weisenfeld and Ott (2011).

We used these as propositions to design our questionnaire, which was structured in three sections. The first section related to the demographics of the participants, specifically age, level of education, field of study/work, experience in the industry, and the interviewees' role in the organization. It is important to clarify the exact nature of the fields of

work included in this questionnaire, since this relates to the participants' involvement within the fine arts market and thus the participants' ownership or otherwise of such items. "Working in insurance" means that the participant is an insurance professional with experience or interest in insuring fine art; "Working in art" means that the respondent is actively involved in the arts sector as an artist or curator of fine arts; "Working in a risk management function" means that the respondent has experience in investing in the fine arts market, either on the participants' own behalf or on behalf of other investors; "Other" means that the respondent works in some other field but nonetheless retains an active interest in fine arts, either as a connoisseur or as an owner; finally, "Working in a combination of functions" simply denotes a mixture of two or more of the above fields.

The second section related to the risks prevalent in the art market. Here, the question required the participants to rate specified risks (Art Damage, Deterioration and loss; Art Fraud; Art Theft; Changes in the Interpretation of Art Value; Fluctuations in the Value of Art; Legal Issues arising from the ownership of artwork, Availability of Information and Changes in the operation of the art market) using a Likert scale ranging from '1' being perceived by the participants as very low risk to '5' being perceived as very high risk. The rest of this section consisted of statements relating to the risks in the art world and these were to be rated by the participants on a Likert scale ranging from '1'—strongly disagree—to '5'—strongly agree.

In the third section we included an open-ended comment box to keep an open mind for any further comments, themes, additions, or clarifications by the participants, which might have been missed in our preliminary study.

### 3.2. The Sample

To gather the participants, we used non-probability purposive and snowball sampling, since we started off with people we knew were knowledgeable about the subject and who believed they could contribute to this study and then with contacts and leads that they suggested (Etikan et al. 2016). In some cases, we administered the survey online using e-communication systems such as 'Zoom', 'MS Teams', and 'Skype', as well as the telephone. For the rest and in the main analysis, we distributed the survey using an online application 'Qualtrics XM', which was administered using social media such as 'Facebook' and 'LinkedIn'. We then asked these prospective participants to help us recruit other subjects among their acquaintances to participate in the study (Ghaljaie et al. 2017; Guest et al. 2006; Mason 2010; Morse 1995; Saunders et al. 2007).

Given the prevalence of fine art in Malta, it is critical to gather an appropriate sample in order to ensure that there is enough statistical power to detect differences in risk perceptions, both in aggregate and across demographics, thereby minimizing the possibility of rejecting false null hypotheses (Type II errors). Assuming a 5% margin of error to ensure that responses broadly reflect those of the population, together with a sampling confidence level of 95%, we determined that a sample size of 384 participants was needed (Creative Research Systems 2020). In total, we collected 465 valid responses from participants who felt they could contribute to this study since they were in one way or another connected to, employed in the area of, or knowledgeable about the subject of this study.

### 3.3. Data Analysis

The respondents' qualitative data were analyzed using the thematic analysis as suggested by Braun and Clarke (2006), while the respondents' quantitative data were subjected to statistical analysis, specifically the non-parametric ANOVA tests: Friedman (distribution of ranks) to answer RQ1 and Kruskal–Wallis (mean of ranks) to answer RQ2.

The thematic analysis approach was used to analyze the open-ended comments/ responses made by participants and the literature. This was achieved by identifying articles and research that was relevant to the study and, once analyzed, we organized them into themes of data that were cohesive. The thematic analysis allowed us to focus our research and provided a clear path to follow.

We used the Friedman Test to determine if there were differences, which are statistically significant between the distributions of three or more related groups when making use of Likert scales. A mean score was generated for each question, which ranged from 1 to 5, in order to determine how respondents rated the risks and statements provided in each section. A mean score close to 1 indicated that the respondent rated the risks and statements as being very low risk, while a mean score close to 5 indicated that the respondent rated the risks and statements as being very high risk.

We use the Kruskal–Wallis test to assess whether there is a significant difference in the fine arts risk perception ranking across a number of demographic variables, including age, occupation, experience, and educational background. We focused on these demographics since, as mentioned earlier, prior studies have shown a clear link between these variables and risk perceptions across a wide variety of domains e.g., (Savage 1993; Sund et al. 2017; Weisenfeld and Ott 2011). The age variable is organized in specific intervals in order to allow for comparisons across age groups while allowing for non-linearities (Andrade 2017), while the education categories are a condensed version of those prescribed by the 2011 International Standard Classification of Education (ISCED) (UNESCO 2012).

## 4. Analysis and Results

### 4.1. Participant Demographics

The sample comprised of a total of 465 valid responses from participants whose grouping with regards to age, education level, occupation, experience, and position are shown in the Tables 1–5 below.

**Table 1.** Age.

|  |  | Frequency | Percent | Valid Percent | Cumulative Percent |
|---|---|---|---|---|---|
|  | 18–24 years | 118 | 25.4 | 25.4 | 25.4 |
|  | 25–34 years | 98 | 21.1 | 21.1 | 46.5 |
|  | 35–44 years | 85 | 18.3 | 18.3 | 64.7 |
| Valid | 45–54 years | 91 | 19.6 | 19.6 | 84.3 |
|  | 55–64 years | 58 | 12.5 | 12.5 | 96.8 |
|  | 65+ | 15 | 3.2 | 3.2 | 100.0 |
|  | Total | 465 | 100.0 | 100.0 |  |

Source: authors' compilation.

**Table 2.** Highest level of education.

|  |  | Frequency | Percent | Valid Percent | Cumulative Percent |
|---|---|---|---|---|---|
|  | Secondary | 24 | 5.2 | 5.2 | 5.2 |
|  | Post-Secondary | 75 | 16.1 | 16.1 | 21.3 |
|  | Undergraduate | 108 | 23.2 | 23.2 | 44.5 |
| Valid | Postgraduate | 234 | 50.3 | 50.3 | 94.8 |
|  | Doctorate and Above | 24 | 5.2 | 5.2 | 100.0 |
|  | Total | 465 | 100.0 | 100.0 |  |

Source: authors' compilation.

As can be noted from the above Tables 1–5, although the largest number of respondents are between the ages of 18 and 24 (118), the other age groups are also well represented. However, there are less valid responses in the age groups of 55 and above. The majority of these participants hold an undergraduate (108) or postgraduate (234) qualification. Additionally, although the majority of participants (259) do not work in either insurance, art, or risk management but work in areas related to fine arts, they felt that they could contribute to the study since they had knowledge of the subject. Moreover, the largest number of respondents (176) worked in the industry for 5 years or less and 128 respondents worked in the industry for 21 years or more.

**Table 3.** Occupation/field of study.

|  |  | Frequency | Percent | Valid Percent | Cumulative Percent |
|---|---|---|---|---|---|
| Valid | Working in Insurance | 125 | 26.9 | 26.9 | 26.9 |
|  | Working in Art | 55 | 11.8 | 11.8 | 38.7 |
|  | Working in a Risk Management Function | 11 | 2.4 | 2.4 | 41.1 |
|  | Other | 259 | 55.7 | 55.7 | 96.8 |
|  | Working in a Combination of Functions | 15 | 3.2 | 3.2 | 100.0 |
|  | Total | 465 | 100.0 | 100.0 |  |

Source: authors' compilation.

**Table 4.** Experience in the industry.

|  |  | Frequency | Percent | Valid Percent | Cumulative Percent |
|---|---|---|---|---|---|
| Valid | 0–5 years | 176 | 37.8 | 37.8 | 37.8 |
|  | 6–10 years | 54 | 11.6 | 11.6 | 49.5 |
|  | 11–15 years | 58 | 12.5 | 12.5 | 61.9 |
|  | 16–20 years | 47 | 10.1 | 10.1 | 72.0 |
|  | 21 years+ | 128 | 27.5 | 27.5 | 99.6 |
|  | Does not want to say | 2 | 0.4 | 0.4 | 100.0 |
|  | Total | 465 | 100.0 | 100.0 |  |

Source: authors' compilation.

**Table 5.** Role/position in the organization.

|  |  | Frequency | Percent | Valid Percent | Cumulative Percent |
|---|---|---|---|---|---|
| Valid | Senior Management | 143 | 30.8 | 30.8 | 30.8 |
|  | Middle Management | 145 | 31.2 | 31.2 | 61.9 |
|  | Clerical | 49 | 10.5 | 10.5 | 72.5 |
|  | Freelance Consultant | 11 | 2.4 | 2.4 | 74.8 |
|  | Student | 43 | 9.2 | 9.2 | 84.1 |
|  | Artist | 12 | 2.6 | 2.6 | 86.7 |
|  | Other related functions | 49 | 10.5 | 10.5 | 97.2 |
|  | Does not Want to Say or Unemployed | 13 | 2.8 | 2.8 | 100.0 |
|  | Total | 465 | 100.0 | 100.0 |  |

Source: authors' compilation.

### 4.2. Risks in the Art Market

Here we sought to establish how participants perceived the risks that are prevalent in the art market (RQ1). The Friedman test on the perceived Art Risks, resulted in a level of significance of less than 0.05 ($p < 0.05$). Therefore, this illustrates that the same respondents rate the risks differently. Table 6 shows the mean rank for each of the presented risks. These illustrate the participants' perception of riskiness level of each risk. The highest risk listed at the top of the list, and decreasing in risk severity as you move down the list.

**Table 6.** Risk perception ranking.

| Perceived Risk | Mean Rank |
|---|---|
| Art damage, deterioration, and loss of art | 5.60 |
| Art fraud and forgery | 5.14 |
| Art theft | 5.08 |
| Changes in the interpretation of art value | 4.42 |
| Fluctuation in the value of art | 4.35 |
| Legal issues arising from the ownership of artworks | 4.03 |
| Availability of information | 3.92 |
| Changes in the operation of the art market | 3.46 |

Source: authors' compilation.

### 4.3. Relationship between Demographic Variables and Art Risk

We used the Kruskal–Wallis Test to analyze the risks against the demographic variables (RQ2).

Table 7 shows a statistically significant difference in the perception of risks derived from "Art Fraud and Forgery" and "Legal issues arising from the ownership art works" based on the age of the participant (*p*-values < 0.05). The age bracket between 35 and 44 years ranked this risk the highest (Mean Rank = 260.08 and 253.87, respectively) with the younger age bracket, 18–24 years, ranking the risk of "Art Fraud and Forgery", and the bracket of 45–54 years ranking the risk of "Legal issues arising from the ownership art works" the lowest (Mean Rank = 207.52 and 187.91, respectively). We further analyze the differences across age groups in the pairwise post-hoc analysis presented in Tables 8 and 9, respectively, which seek to determine whether perceptions regarding both art fraud and forgery and legal issues differ according to age groups. As seen below, for art forgery and fraud (Table 8) the key statistically significant difference lies between the 18 to 24 and the 35 to 44 age bracket (adj. sig. 0.04), whereas for legal issues arising from ownership of art works (Table 9) we find statistically significant differences across the 45–54 and 18–24 age brackets (adj. sig. = 0.008) and the 45–54 and 35–44 age brackets (adj. sig. = 0.005).

**Table 7.** Effects of age on art risk perception test statistics [a,b].

| | Art Fraud and Forgery | Art Theft | Art Damage, Deterioration and Loss of Art | Fluctuation in the Value of Art | Legal Issues Arising from the Ownership Art Works | Changes in the Operation of the Art Market | Changes in the Interpretation of Art Value | Availability of Information |
|---|---|---|---|---|---|---|---|---|
| K-W | 9.541 ** | 3.962 | 6.367 | 2.935 | 18.810 ** | 6.973 | 8.646 | 6.324 |
| df | 4 | 4 | 4 | 4 | 4 | 4 | 4 | 4 |
| Asymp. Sig. | 0.049 | 0.411 | 0.173 | 0.569 | 0.001 | 0.137 | 0.071 | 0.176 |

[a]. Kruskal–Wallis Test [b]. Grouping variable: age. Source: authors' compilation. ** denotes that the result is statistically significant at the 5% level.

Table 10 shows a statistically significant difference in the perception of risks derived from 'Art Fraud and Forgery', 'Art Damage, Deterioration and Loss of Art', and 'Fluctuation in the value of art' and 'Availability of information', based on the level of education of the participant (*p*-values < 0.05). The participants holding a doctorate level or above ranked the risk of 'Art Fraud and Forgery', 'Art Damage, Deterioration and Loss of Art', and 'Fluctuation in the value of art' the highest (Mean Rank = 289.92, 254.38 and 286.85 respectively), and those holding a postgraduate degree ranked the risk of 'Availability of information' the highest (Mean Rank = 249.32). While the lowest ranking of the risk of Art Fraud and Forgery', 'Art Damage, Deterioration and Loss of Art', 'Fluctuation in the value of art', and 'Availability of information' was perceived by those holding a secondary level of education (Mean Rank = 165.67, 163.19, 178.31 and 174.83, respectively).

**Table 8.** Post hoc analysis—art fraud and forgery.

| Sample 1-Sample 2 | Test Statistic | Std. Error | Std. Test Statistic | Sig. | Adj. Sig. [a] |
|---|---|---|---|---|---|
| 18–24 years-25–34 years | −11.581 | 17.393 | −0.666 | 0.506 | 1.000 |
| 18–24 years-45–54 years | −16.228 | 17.755 | −0.914 | 0.361 | 1.000 |
| 18–24 years-65+ | −17.735 | 34.886 | −0.508 | 0.611 | 1.000 |
| 18–24 years-55–64 years | −19.740 | 20.409 | −0.967 | 0.333 | 1.000 |
| 18–24 years-35–44 years | −54.437 | 18.106 | −3.007 | 0.003 | 0.040 |
| 25–34 years-45–54 years | −4.647 | 18.527 | −0.251 | 0.802 | 1.000 |
| 25–34 years-65+ | −6.154 | 35.285 | −0.174 | 0.862 | 1.000 |
| 25–34 years-55–64 years | −8.158 | 21.084 | −0.387 | 0.699 | 1.000 |
| 25–34 years-35–44 years | −42.856 | 18.863 | −2.272 | 0.023 | 0.346 |
| 45–54 years-65+ | −1.507 | 35.465 | −0.042 | 0.966 | 1.000 |
| 45–54 years-55–64 years | −3.512 | 21.383 | −0.164 | 0.870 | 1.000 |
| 45–54 years-35–44 years | 38.209 | 19.197 | 1.990 | 0.047 | 0.698 |
| 65+-55-64 years | 2.005 | 36.865 | 0.054 | 0.957 | 1.000 |
| 65+-35–44 years | 36.702 | 35.642 | 1.030 | 0.303 | 1.000 |
| 55–64 years-35–44 years | 34.697 | 21.675 | 1.601 | 0.109 | 1.000 |

Each row tests the null hypothesis that the Sample 1 and Sample 2 distributions are the same. Asymptotic significances (2-sided tests) are displayed. The significance level is 0.05. [a] Significance values have been adjusted by the Bonferroni correction for multiple tests.

**Table 9.** Post hoc analysis—Legal issues arising from ownership of art works.

| Sample 1-Sample 2 | Test Statistic | Std. Error | Std. Test Statistic | Sig. | Adj. Sig. [a] |
|---|---|---|---|---|---|
| 45–54 years-55–64 years | −14.876 | 21.242 | −0.700 | 0.484 | 1.000 |
| 45–54 years-25–34 years | 37.340 | 18.405 | 2.029 | 0.042 | 0.637 |
| 45–54 years-65+ | −45.253 | 35.230 | −1.284 | 0.199 | 1.000 |
| 45–54 years-18–24 Years | 61.072 | 17.638 | 3.463 | 0.001 | 0.008 |
| 45–54 years-35–44 years | 68.404 | 19.070 | 3.587 | 0.000 | 0.005 |
| 55–64 years-25–34 years | 22.463 | 20.944 | 1.073 | 0.283 | 1.000 |
| 55–64 years-65+ | −30.377 | 36.621 | −0.829 | 0.407 | 1.000 |
| 55–64 years-18–24 years | 46.196 | 20.274 | 2.279 | 0.023 | 0.340 |
| 55–64 years-35–44 years | 53.528 | 21.532 | 2.486 | 0.013 | 0.194 |
| 25–34 years-65+ | −7.914 | 35.052 | −0.226 | 0.821 | 1.000 |
| 25–34 years-18–24 years | 23.733 | 17.278 | 1.374 | 0.170 | 1.000 |
| 25–34 years-35–44 years | −31.065 | 18.739 | −1.658 | 0.097 | 1.000 |
| 65+-18–24 years | 15.819 | 34.655 | 0.456 | 0.648 | 1.000 |
| 65+-35–44 years | 23.151 | 35.406 | 0.654 | 0.513 | 1.000 |
| 18–24 years-35–44 years | −7.332 | 17.986 | −0.408 | 0.684 | 1.000 |

Each row tests the null hypothesis that the Sample 1 and Sample 2 distributions are the same. Asymptotic significances (2-sided tests) are displayed. The significance level is 0.050. [a] Significance values have been adjusted by the Bonferroni correction for multiple tests.

**Table 10.** Effects of level of education on art risk perception test statistics [a,b].

| | Art Fraud and Forgery | Art Theft | Art Damage, Deterioration, and Loss of Art | Fluctuation in the Value of Art | Legal Issues Arising from the Ownership Art Works | Changes in the Operation of the Art Market | Changes in the Interpretation of Art Value | Availability of Information |
|---|---|---|---|---|---|---|---|---|
| K-W | 19.700 ** | 8.095 | 18.228 ** | 11.365 ** | 8.060 | 3.085 | 2.982 | 11.537 ** |
| df | 4 | 4 | 4 | 4 | 4 | 4 | 4 | 4 |
| Asymp. Sig. | 0.001 | 0.088 | 0.001 | 0.023 | 0.089 | 0.544 | 0.561 | 0.021 |

[a] Kruskal–Wallis Test. [b] Grouping variable: level of education. Source: authors' compilation ** denotes that the result is statistically significant at the 5% level.

We can further probe these results using the post-hoc analysis presented in Tables 11–14 for each specific risk perception factor. As seen below, for Art fraud and forgery (Table 11) the key differences lie across the secondary and postgraduate educa-

tional groups (adj. sig. = 0.023) and across the secondary-doctorate groups (adj. sig. = 0.007). For Art damage, deterioration, and loss (Table 12), the main differences lie across secondary and postgraduates (adj. sig. = 0.029), secondary and undergraduates (adj. sig. = 0.037), and post-secondary and postgraduates (adj. sig. = 0.026). When it comes to Fluctuation in the value of art (Table 13), the main differences lie across secondary and doctorates (adj. sig. = 0.03) only, while for Availability of information (Table 14) there are no statistically significant differences across our pairs of educational groups.

**Table 11.** Post hoc analysis—art fraud and forgery.

| Sample 1-Sample 2 | Test Statistic | Std. Error | Std. Test Statistic | Sig. | Adj. Sig. [a] |
|---|---|---|---|---|---|
| Secondary-Post Secondary | −40.867 | 29.847 | −1.369 | 0.171 | 1.000 |
| Secondary-Undergraduate | −53.519 | 28.720 | −1.863 | 0.062 | 0.624 |
| Secondary-Postgraduate | −83.261 | 27.278 | −3.052 | 0.002 | 0.023 |
| Secondary-Doctorate and Above | −124.250 | 36.739 | −3.382 | 0.001 | 0.007 |
| Post-Secondary-Undergraduate | −12.652 | 19.129 | −0.661 | 0.508 | 1.000 |
| Post-Secondary-Postgraduate | −42.394 | 16.887 | −2.510 | 0.012 | 0.121 |
| Post-Secondary-Doctorate and Above | −83.383 | 29.847 | −2.794 | 0.005 | 0.052 |
| Undergraduate-Postgraduate | −29.742 | 14.805 | −2.009 | 0.045 | 0.445 |
| Undergraduate-Doctorate and Above | −70.731 | 28.720 | −2.463 | 0.014 | 0.138 |
| Postgraduate-Doctorate and Above | −40.989 | 27.278 | −1.503 | 0.133 | 1.000 |

Each row tests the null hypothesis that the Sample 1 and Sample 2 distributions are the same. Asymptotic significances (2-sided tests) are displayed. The significance level is 0.050. [a] Significance values have been adjusted by the Bonferroni correction for multiple tests.

**Table 12.** Post hoc analysis—art damage, deterioration and loss.

| Sample 1-Sample 2 | Test Statistic | Std. Error | Std. Test Statistic | Sig. | Adj. Sig. [a] |
|---|---|---|---|---|---|
| Secondary-Post Secondary | −30.206 | 29.880 | −1.011 | 0.312 | 1.000 |
| Secondary-Postgraduate | −81.199 | 27.309 | −2.973 | 0.003 | 0.029 |
| Secondary-Undergraduate | −83.410 | 28.752 | −2.901 | 0.004 | 0.037 |
| Secondary-Doctorate and Above | −91.188 | 36.780 | −2.479 | 0.013 | 0.132 |
| Post-Secondary-Postgraduate | −50.993 | 16.906 | −3.016 | 0.003 | 0.026 |
| Post-Secondary-Undergraduate | −53.204 | 19.151 | −2.778 | 0.005 | 0.055 |
| Post-Secondary-Doctorate and Above | −60.982 | 29.880 | −2.041 | 0.041 | 0.413 |
| Postgraduate-Undergraduate | 2.210 | 14.822 | 0.149 | 0.881 | 1.000 |
| Postgraduate-Doctorate and Above | −9.988 | 27.309 | −0.366 | 0.715 | 1.000 |
| Undergraduate-Doctorate and Above | −7.778 | 28.752 | −0.271 | 0.787 | 1.000 |

Each row tests the null hypothesis that the Sample 1 and Sample 2 distributions are the same. Asymptotic significances (2-sided tests) are displayed. The significance level is 0.050. [a] Significance values have been adjusted by the Bonferroni correction for multiple tests.

**Table 13.** Post hoc analysis—fluctuation in the value of art.

| Sample 1-Sample 2 | Test Statistic | Std. Error | Std. Test Statistic | Sig. | Adj. Sig. [a] |
|---|---|---|---|---|---|
| Secondary-Post Secondary | −37.961 | 29.733 | −1.277 | 0.202 | 1.000 |
| Secondary-Undergraduate | −48.488 | 28.610 | −1.695 | 0.090 | 0.901 |
| Secondary-Postgraduate | −62.995 | 27.174 | −2.318 | 0.020 | 0.204 |
| Secondary-Doctorate and Above | −108.542 | 36.599 | −2.966 | 0.003 | 0.030 |
| Post-Secondary-Undergraduate | −10.528 | 19.056 | −0.552 | 0.581 | 1.000 |
| Post-Secondary-Postgraduate | −25.034 | 16.823 | −1.488 | 0.137 | 1.000 |
| Post-Secondary-Doctorate and Above | −70.581 | 29.733 | −2.374 | 0.018 | 0.176 |
| Undergraduate-Postgraduate | −14.507 | 14.749 | −0.984 | 0.325 | 1.000 |
| Undergraduate-Doctorate and Above | −60.053 | 28.610 | −2.099 | 0.036 | 0.358 |
| Postgraduate-Doctorate and Above | −45.546 | 27.174 | −1.676 | 0.094 | 0.937 |

Each row tests the null hypothesis that the Sample 1 and Sample 2 distributions are the same. Asymptotic significances (2-sided tests) are displayed. The significance level is 0.050. [a] Significance values have been adjusted by the Bonferroni correction for multiple tests.

**Table 14.** Post hoc Analysis—Availability of information.

| Sample 1-Sample 2 | Test Statistic | Std. Error | Std. Test Statistic | Sig. | Adj. Sig. [a] |
|---|---|---|---|---|---|
| Secondary-Post Secondary | −35.713 | 30.027 | −1.189 | 0.234 | 1.000 |
| Secondary-Undergraduate | −50.083 | 28.894 | −1.733 | 0.083 | 0.830 |
| Secondary-Doctorate and Above | −63.771 | 36.961 | −1.725 | 0.084 | 0.845 |
| Secondary-Postgraduate | −74.485 | 27.443 | −2.714 | 0.007 | 0.066 |
| Post-Secondary-Undergraduate | −14.370 | 19.245 | −0.747 | 0.455 | 1.000 |
| Post-Secondary-Doctorate and Above | −28.058 | 30.027 | −0.934 | 0.350 | 1.000 |
| Post-Secondary-Postgraduate | −38.772 | 16.989 | −2.282 | 0.022 | 0.225 |
| Undergraduate-Doctorate and Above | −13.688 | 28.894 | −0.474 | 0.636 | 1.000 |
| Undergraduate-Postgraduate | −24.402 | 14.895 | −1.638 | 0.101 | 1.000 |
| Doctorate and Above-Postgraduate | 10.714 | 27.443 | 0.390 | 0.696 | 1.000 |

Each row tests the null hypothesis that the Sample 1 and Sample 2 distributions are the same. Asymptotic significances (2-sided tests) are displayed. The significance level is 0.050. [a] Significance values have been adjusted by the Bonferroni correction for multiple tests.

Table 15 shows a statistically significant difference in the perception of risks derived from 'Fluctuation in the value of art', 'Changes in the interpretation of art value', and 'Availability of information', based on the participant Occupation/Field of Study (*p*-values are 0.036, 0.006, and 0.001, respectively). The participants working in a combination of functions ranked the risk of 'Fluctuation in the value of art' and 'Availability of information' the highest (Mean Rank = 3.400 for both). Participants working in Art ranked the risk of 'Changes in the interpretation of art value' the highest (Mean Rank = 2.9273). On the other hand, those working in a risk management function ranked the risk of 'Fluctuation in the value of art' and 'Availability of information' lowest (Mean Rank = 2.5455 and 2.3636 respectively). While the lowest ranking of the risk of 'Changes in the interpretation of art value' was perceived by those w (Mean Rank = 2.7600).

We also present the post-hoc analysis for each of the three risk perception factors discussed above in Tables 16–18. When it comes to Fluctuation in the value of art (Table 16), the statistically significant differences are observed across working in risk management versus working in art (adj. sig. = 0.045), for Changes in the interpretation of art value (Table 17) the key difference is between those working in risk management and those who work in a combination of functions (adj. sig. = 0.044), and for Availability of information (Table 18) the main differences lie across those working in other sectors and those within the insurance sector (adj. sig. = 0.006).

**Table 15.** Effects of occupation/field of study on art risk perception.

| Test Statistics [a,b] | | | | | | | | |
|---|---|---|---|---|---|---|---|---|
| | Art Fraud and Forgery | Art Theft | Art Damage, Deterioration, and Loss of Art | Fluctuation in the Value of Art | Legal Issues Arising from the Ownership art Works | Changes in the Operation of the art Market | Changes in the Interpretation of art Value | Availability of Information |
| K-W | 0.788 | 4.898 | 7.874 | 10.286 ** | 1.843 | 1.920 | 14.567 ** | 18.095 ** |
| df | 4 | 4 | 4 | 4 | 4 | 4 | 4 | 4 |
| Asymp. Sig. | 0.940 | 0.298 | 0.096 | 0.036 | 0.765 | 0.750 | 0.006 | 0.001 |

[a] Kruskal–Wallis Test [b] Grouping variable: occupation/field of study. Source: authors' compilation. ** denotes that the result is statistically-significant at the 5% level.

**Table 16.** Post hoc analysis—fluctuation in the value of art.

| Sample 1-Sample 2 | Test Statistic | Std. Error | Std. Test Statistic | Sig. | Adj. Sig. [a] |
|---|---|---|---|---|---|
| Working in a Risk Management Function-Working in Insurance | 84.595 | 39.872 | 2.122 | 0.034 | 0.339 |
| Working in a Risk Management Function-Other | −86.594 | 39.029 | −2.219 | 0.027 | 0.265 |
| Working in a Risk Management Function-Working in Art | 118.900 | 41.874 | 2.839 | 0.005 | 0.045 |
| Working in a Risk Management Function-Working in a Combination of Functions | −129.927 | 50.327 | −2.582 | 0.010 | 0.098 |
| Working in Insurance-Other | −1.999 | 13.808 | −0.145 | 0.885 | 1.000 |
| Working in Insurance-Working in Art | −34.305 | 20.514 | −1.672 | 0.094 | 0.945 |
| Working in Insurance-Working in a Combination of Functions | −45.332 | 34.643 | −1.309 | 0.191 | 1.000 |
| Other-Working in Art | 32.306 | 18.823 | 1.716 | 0.086 | 0.861 |
| Other-Working in a Combination of Functions | −43.333 | 33.669 | −1.287 | 0.198 | 1.000 |
| Working in Art-Working in a Combination of Functions | −11.027 | 36.930 | −0.299 | 0.765 | 1.000 |

Each row tests the null hypothesis that the Sample 1 and Sample 2 distributions are the same. Asymptotic significances (2-sided tests) are displayed. The significance level is 0.050. [a] Significance values have been adjusted by the Bonferroni correction for multiple tests.

**Table 17.** Post hoc analysis—changes in the interpretation of art value.

| Sample 1-Sample 2 | Test Statistic | Std. Error | Std. Test Statistic | Sig. | Adj. Sig. [a] |
|---|---|---|---|---|---|
| Working in a Risk Management Function-Working in Art | 53.800 | 42.295 | 1.272 | 0.203 | 1.000 |
| Working in a Risk Management Function-Other | −91.062 | 39.421 | −2.310 | 0.021 | 0.209 |
| Working in a Risk Management Function-Working in Insurance | 106.081 | 40.273 | 2.634 | 0.008 | 0.084 |
| Working in a Risk Management Function-Working in a Combination of Functions | −144.839 | 50.832 | −2.849 | 0.004 | 0.044 |
| Working in Art-Other | −37.262 | 19.012 | −1.960 | 0.050 | 0.500 |
| Working in Art-Working in Insurance | 52.281 | 20.720 | 2.523 | 0.012 | 0.116 |
| Working in Art-Working in a Combination of Functions | −-91.039 | 37.301 | −2.441 | 0.015 | 0.147 |
| Other-Working in Insurance | 15.018 | 13.946 | 1.077 | 0.282 | 1.000 |
| Other-Working in a Combination of Functions | −53.777 | 34.007 | −1.581 | 0.114 | 1.000 |
| Working in Insurance-Working in a Combination of Functions | −38.759 | 34.991 | −1.108 | 0.268 | 1.000 |

Each row tests the null hypothesis that the Sample 1 and Sample 2 distributions are the same. Asymptotic significances (2-sided tests) are displayed. The significance level is 0.050. [a] Significance values have been adjusted by the Bonferroni correction for multiple tests.

**Table 18.** Post hoc analysis—availability of information.

| Sample 1-Sample 2 | Test Statistic | Std. Error | Std. Test Statistic | Sig. | Adj. Sig. [a] |
|---|---|---|---|---|---|
| Working in a Risk Management Function-Other | −50.782 | 39.416 | −1.288 | 0.198 | 1.000 |
| Working in a Risk Management Function-Working in Art | 62.636 | 42.289 | 1.481 | 0.139 | 1.000 |
| Working in a Risk Management Function-Working in Insurance | 98.858 | 40.267 | 2.455 | 0.014 | 0.141 |
| Working in a Risk Management Function-Working in a Combination of Functions | −125.552 | 50.825 | −2.470 | 0.014 | 0.135 |
| Other-Working in Art | 11.855 | 19.009 | 0.624 | 0.533 | 1.000 |
| Other-Working in Insurance | 48.077 | 13.944 | 3.448 | 0.001 | 0.006 |
| Other-Working in a Combination of Functions | −74.770 | 34.003 | −2.199 | 0.028 | 0.279 |
| Working in Art-Working in Insurance | 36.222 | 20.717 | 1.748 | 0.080 | 0.804 |
| Working in Art-Working in a Combination of Functions | −62.915 | 37.295 | −1.687 | 0.092 | 0.916 |
| Working in Insurance-Working in a Combination of Functions | −26.693 | 34.986 | −0.763 | 0.445 | 1.000 |

Each row tests the null hypothesis that the Sample 1 and Sample 2 distributions are the same. Asymptotic significances (2-sided tests) are displayed. The significance level is 0.050. [a] Significance values have been adjusted by the Bonferroni correction for multiple tests.

On the other hand, Tables 19 and 20, show no statistically significant difference in the perception of risks based on the participants' years of experience in the industry ($p$-values > 0.05).

**Table 19.** Effects of position in the organization on art risk perception. Test statistics [a,b].

|  | Art fraud and Forgery | Art theft | Art Damage, Deterioration, and Loss of Art | Fluctuation in the Value of Art | Legal Issues Arising from the Ownership Art Works | Changes in the Operation of the Art Market | Changes in the Interpretation of Art Value | Availability of Information |
|---|---|---|---|---|---|---|---|---|
| K-W | 0.045 | 1.251 | 0.032 | 2.274 | 2.472 | 0.817 | 1.057 | 3.109 |
| df | 1 | 1 | 1 | 1 | 1 | 1 | 1 | 1 |
| Asymp. Sig. | 0.831 | 0.263 | 0.859 | 0.132 | 0.116 | 0.366 | 0.304 | 0.078 |

[a] Kruskal–Wallis Test. [b] Grouping variable: role/position in the organization. Source: authors' compilation.

**Table 20.** Effects of years of experience in the industry on art risk perception. Test statistics [a,b].

|  | Art fraud and Forgery | Art Theft | Art Damage, Deterioration, and Loss of Art | Fluctuation in the Value of Art | Legal Issues Arising from the Ownership Art Works | Changes in the Operation of the Art Market | Changes in the Interpretation of Art Value | Availability of Information |
|---|---|---|---|---|---|---|---|---|
| K-W | 4.061 | 2.365 | 4.473 | 4.963 | 8.377 | 0.524 | 6.160 | 4.011 |
| df | 4 | 4 | 4 | 4 | 4 | 4 | 4 | 4 |
| Asymp. Sig. | 0.398 | 0.669 | 0.346 | 0.291 | 0.079 | 0.971 | 0.188 | 0.405 |

[a] Kruskal–Wallis Test. [b] Grouping variable: experience in the industry. Source: authors' compilation.

*4.4. Thematic Analysis on Further Comments*

Some of the respondents (17) continued to emphasize the importance of the risk during the restoration process. They noted that if this is not done properly or professionally, it could severely damage a work of art or devalue it completely. Respondents (eight) continued to argue that buying a work of art for 'investment purposes' rather than for its aesthetic value distorts the valuation of the works of art. Other participants (17) elaborated on the risk of 'forgeries' by including 'imitations', which they insinuate can distort the valuation of fine art. They note that while forgeries might not be worth much, imitated or copied art which is sold as such can have a high value and can be considered as an art in itself. Sellers must, however, be transparent about the piece and give the buyer all information about it and specify that it is an imitation or copied. Additionally, it might be easier to establish the authenticity of some works as opposed to others, since there are organization specializing in authenticating certain artists and thus making it easier to value.

Some respondents (14) mentioned 'transportation or transit risk', of which a piece could be exposed to. An artwork would be at a higher risk of being damaged in transit rather than hung up in a museum or on exhibition. They also mentioned 'storage and environment' in which the artwork is kept. This could include factors such as whether the piece is in a museum or someone's house, which hold different risk levels, or the way in which it is being taken care of. Other respondents (nine) commented about the 'risk of replacement' and noted that art insurance is similar to life insurance, in that it cannot replace the artwork that was insured, but rather provide the damaged party with a sum to the insured. This is in contrast to other policies, whereby, if, for example, there was damage to a building, the damage can be repaired. Some (22) participants also mentioned the risk of fine art being used as a means for 'laundering money'. This today distorts the valuation process even when specialist valuers are involved.

## 5. Conclusions

This paper sought to analyze risk perceptions related to the market for fine art, both in terms of severity as well as the extent to which these perceptions vary according to different demographic variables. It was determined that the highest rated perceived risk was that of "Art damage, deterioration, and loss of art", followed closely by "Art fraud and forgery" and "Art theft". In contrast to this, participants perceived "Changes in the operation of the art market" to offer the lowest risk. All risks were rated as medium, when considering the mean response rate, except for "Art damage, deterioration and loss of art", which was categorized as being high risk. (RQ1).

Findings further revealed that certain demographics had an effect on the way some of the risks were ranked, mainly that (1) 'Age' affected the perceived ranking for 'Audit fraud and forgery' and 'Legal issues arising from the ownership art works' risk; (2) 'Level of education' affected the perceived ranking of 'Art fraud and forgery', 'Art damage, deterioration and loss of art', and 'Fluctuation in the value of art' and 'Availability of information' risk; (3) 'Field of study' affects the perceived ranking of 'Fluctuation in the value of art', 'Changes in the interpretation of art value', and 'Availability of information' risk. These findings broadly reflect results in other studies looking at how risk perceptions across different domains vary according to individual demographics. By contrast, 'Position in the organization' and 'Experience in the industry' had no effect on the perceived ranking of the different risks. (RQ2).

In carrying out this process we also revealed some other perceived risks that might have been missed when analyzing literature or which the respondents wished to emphasize further via their qualitative responses. These relate to the following: (1) The restoration process, (2) Investing through art, (3) Forgeries and imitations, (4) Irreplaceability, (5) Transportation or transit risk, and (6) Money laundering.

At this point, it is worth noting a number of important limitations with the above analysis. Firstly, the data were all derived from self-reported questionnaire responses as opposed to induced or observed behavior, and thus must be treated with some degree of caution. Secondly, and perhaps related to the previous point, we found no significant relationship between occupation and years of experience, two of our demographic variables, and any of the risk perceptions, while certain risk perceptions such as art theft and changes in the operation of the art market cannot be explained by any of the demographic variables used. This points towards a need to potentially expand the scope of analysis beyond demographic variables into other socio-economic or behavioral characteristics in order to adequately explain and account for variation in risk perceptions, at least when it comes to future studies related to fine art and insurance.

Nonetheless, these results provide the industry with an insight of risks that need to be addressed by holders of art, as well as considerations for policymakers of fine art insurance policies and provide a basis for which such a product/s can be considered to be introduced in Malta. The perceived risks as seen by the sample taken from the Maltese population, as well as people working in the art and insurance industries, provide us with an idea of what needs to be taken into account when and if offering fine art insurance. In fact, the results indicate that, based on risk perceptions, the individuals who would be most open to fine art insurance given their awareness of risks are in the 35–44 age bracket, typically with a postgraduate (or higher) level of education. Thus, the design and marketing of a fine arts insurance policy should be aimed at this segment of the market, taking into account their specific risk perceptions as well as expenditure propensities and abilities for related premiums. This will in turn assist in providing adequate protection for fine art, based on the highest-ranked risks like art damage, deterioration and losses, coupled with fraud, and create a new revenue stream for insurance companies.

**Author Contributions:** Conceptualization, methodology, software, L.P. and S.G.; validation, S.G., J.V.S., and I.R.; formal analysis, L.P., J.V.S., and S.G.; investigation, L.P.; resources, L.P.; data curation, S.G. and L.P.; writing—original draft preparation, L.P.; writing—review and editing, S.G., J.V.S., and I.R.; visualization, I.R.; supervision, S.G.; project administration, L.P. All authors have read and agreed to the published version of the manuscript.

**Funding:** This research received no external funding.

**Institutional Review Board Statement:** The study was conducted according to the guidelines of the Declaration of Helsinki, and approved by the faculty Ethics Committee of University of Malta faculty of Economics, Management and Accountancy Ethics Committee. MSD2080, Malta (Form ID 4739_30032020) on the 30 March 2020.

**Informed Consent Statement:** Not applicable.

**Data Availability Statement:** The data presented in this study are available on request from the corresponding author. The data are not publicly available due to ethical concerns.

**Acknowledgments:** The article is based on an unpublished thesis in partial fulfilment of Luke Pavia's in Insurance and Risk Management at the University of Malta, supervised by Simon Grima both authors of this article.

**Conflicts of Interest:** The authors declare no conflict of interest.

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
