# Peer review of "Fine Art Insurance Policies and Risk Perceptions: The Case of Malta"

_jrfm, doi:10.3390/jrfm14020066_

Round 1

Reviewer 1 Report

Review response

The paper is interesting and explores the relatively rare problem. However, it is very long and Authors very often miss their main problem in focus. Thus, I recommend Authors to consider shortening some parts/aspects (some are not needed and only make some ‘noise’ and disturb a reader in following the main problem)

My overall impression is that you tried to put too many eggs in one basket. It is really hard to understand (a) what was the subject of your survey and why, (b) why you have asked the given set of questions in the context of fine art risk perception, (c) why these particular demographics were relevant for you, (d) how it all corresponds to the problem of underwriting.

In general, I recommend not to refer to the underwriting issues (this is not the point in your study) and deeply reconsider your research question and the whole presentation of the statistical analysis. It raises so many ambiguities and is very poorly interpreted (some tests are not completed in fact, I mean the post-hoc analysis in KW).

I suggest to reconsider the presentation of these results in two separate papers: (1) on risk perception and demographics, (2) general analysis of the risk-relevant factors.

I whish you good luck with this study!

Below I provide some detailed suggestions on how you can improve your work before resubmission:

Introduction

Authors subdivided the introduction into 3 parts, in fact. In my opinion, section 2 (aim and objective) and section 3 (why Malta) should be integrated with the current introduction (section 1). Please note that in section 2 (aim and objective) you actually explain why Malta as well. Moreover, the reference to Notre-Dame example should be reconsidered to highlight better the contribution and value of this study. Currently, you tried, but in fact the way in which you refer to Notre-Dame case states something slightly different than approached in this study…

Finally, the introduction needs to clearly explain the purpose of the work, given the existing research gap and by this highlighting the contribution of your work. Although you try to collaborate these items it still needs substantive work.

My suggestion is to shorten the reference to Malta case in the introduction. You may use some parts of this text later on, while discussing research design.

Section 3 (which should be 4 in fact) – research questions. The research questions should be presented within the presentation of research design and method.

Section 4 – Literature review.

This section is very problematic. It is very long, there are aspects that are far from the main course of this paper. Do you really need to explain so widely what is art, fine art, definition of fraud and forgery? Please remember that your study is on insurance, so the reader expect that you will go straight to the point, rather than discussing so widely some well-established defintions (at least in the common understanding), as well as the historical contexts (eg. The text on the committed acts of theft in the past…)

Your literature review should refer only to a) valuation of art for the purposes of insurance (and please shorten the reference to social aspect!) b) the insurance of art (but please do not begin with what is insurance and the types of insurance contracts (life insurance in particular)

Instead, your literature review is missing an overview of prior works that discussed the problem of a) perception of the relevance of fine art insurance and b) the problems of underwriting in fine art insurance. In this respect, the discussion on commercial value (section 4.5.1) is incomplete – there are no reference to the literature on underwriting and the related problems while underwriting based on market value.

Section 5 Research methods

First, this section goes beyond the presentation of research method, thus it should be titled research DESIGN and methods.

The fact that you plan a survey (and what is overall the scope of this survey) should be clearly communicated in the introduction as well!)

In lines 492-494 you stated that you refer to the risks based on the literature, but nothing was previously discussed within literature review!

Reference to Friedman test – in my opinion it would be enough to communicate that this test is compares the distribution of scores (the whole explanation of the hypotheses and the p-values is definitely not needed). Similarly, the reference to Kruskall-Wallis test. All these efforts could be exchanged by one sentence (which is communicative for researchres in this field: Due to the nature of data (survey-based) we relied on non-parametric ANOVA tests: Friedman (distribution of ranks) and Kruskal-Wallis (mean of ranks).

This section is missing the presentation of your research questions. Please note that the RQs shall correspond nicely with the further presentation of the results and the related discussion (problem by problem, not test by test)

Section 6. Analysis and Results

The presentation of demographics could be more communicative if you present some graphics, rather than discuss the percentage structure. From methodical point of view, this is not clear why and how you considered the age of your respondents, the level of education, insurance-background or where they work. This all should be related to prior literature (why these items are relevant for your studies).

This is even more important given the further extensive analysis of demographics and art risk perception.

I have a problem with how you interpreted the results of Friedmant test (distrubutions). For me, you should first present the table 1b, then fig 1, and then the results of the test. Please note that the hypothesis you formed is far from the RQ you placed… You should highlight which RQ is answered by verification of this hypothesis.

Now I will refer to my former doubts on the lack of explanation on why you refer to the given demographic characteristics

  • Age – do the age intervals in your study are motivated by prior studies on risk perception? Was it e.g. confirmed that the younger people are more risk averse? Please also note that you actually don’t answer the core question – which “age” group has evaluated the risk as higher/lower. That is the most interesting point here. You need to demonstrate the results of post-hoc tests and ideally the mean ranks of KW test. Otherwise, this is very superficial what you did. Instead of such a wide presentation of means scores broken by the demographics, the post-hoc tests should be presented here.
  • The same refers to your further statistics. I also noted that while you were more communicative with age, the presentation of other demographics is insufficient (you only place a table… with one sentence and this sentence is actually not needed for the academics in this field, as we all know which statistics were significant)

I am not sure if this is a right research procedure to mix EFA in the way you provided in this work. EFA is used if you don’t know what you may expect… Then you may apply other tests (e.g. KW – which you actually did later on again!). Also, why Cronbach-Alfa is presented at the very end? We typically explain this at the beginning, only to justify the ‘quality’ of the survey questions.

Honestly, I got lost on what and why you do… I cannot track the main problems (risk perception, underwriting) and the RQs you placed…

The table 24b comes out of nowhere, I have no clue why these particular items are used as the hallmarks of the perceived risk? And how it refers to the risk perception items discussed above?

Finally, where is the underwriting? How your analysis refers to the problem of underwriting (the market value of fine art?)

Section 7 – conclusions

The conclusions could be written much better. Currently, this is mainly the summary of the results (written in similar style as it was done previously).

Author Response

Thank you very much for your comments and recommendations, which have been much appreciated and duly noted. This has made our paper much better. Answers and clarifications have been provided in blue below your evaluation, for ease of reading.

Hope this is to your satisfaction.

Kind regards,

The authors

The paper is interesting and explores the relatively rare problem. However, it is very long and Authors very often miss their main problem in focus. Thus, I recommend Authors to consider shortening some parts/aspects (some are not needed and only make some ‘noise’ and disturb a reader in following the main problem).

Following your recommendation, we have shortened the paper to take off as much noise as possible and concentrate on the main problem studied in the paper Vide RQ1 and RQ2.

RQ1-   What are the risks that need to be addressed when holding art and which risks are perceived as being riskiest?

RQ2-   How do demographic variables affect people’s perception of art risks and art insurance?

My overall impression is that you tried to put too many eggs in one basket. It is really hard to understand (a) what was the subject of your survey and why, (b) why you have asked the given set of questions in the context of fine art risk perception, (c) why these particular demographics were relevant for you, (d) how it all corresponds to the problem of underwriting.

We have compressed this section to explain our main aim better. The formulation of the aim is clarified: we aim to identify the risks that need to be addressed when holding fine art and determine which is perceived as being the riskiest. Moreover, in doing this, we analyze whether the risk perception is influenced by demographic variables related to the fine art owners. Therefore, we have reduced the number of research questions to two:

RQ1-   What are the risks that need to be addressed when holding art and which risks are perceived as being riskiest?

This would help in determining the possible market demand for an insurance policy on fine art and therefore its pricing/perceived value.

RQ2-   How do demographic variables affect people’s perception of art risks and art insurance?

This would help to determine whether the risk perception is influenced by demographic variables.

We have chosen these particular Demographics in RQ2- since as noted in previous studies (Bezzina et al. 2012 and 2014) on other kinds of contracts and perceptions of risk they are the variables that more generally have an effect on perceptions of risk and therefore on the demand which will have an effect on the value of the risk and therefore on the demand which will have an effect on the potential entry of Insurers in this market and the value of the risk.

Knowing the demographics of a country and relating this to perceived risks of fine arts will enable Insurers to determine the potential clientele, the value these clientele give to the fine art and in turn the premium to be charged for such a policy.

Reference to underwriting was taken off as suggested.

In general, I recommend not to refer to the underwriting issues (this is not the point in your study) and deeply reconsider your research question and the whole presentation of the statistical analysis. It raises so many ambiguities and is very poorly interpreted (some tests are not completed in fact, I mean the post-hoc analysis in KW).

Reference to underwriting was taken off as suggested. We interpreted the mean ranks of KW.

I suggest to reconsider the presentation of these results in two separate papers: (1) on risk perception and demographics, (2) general analysis of the risk-relevant factors.

As recommended, we focus this paper on RQ1 and RQ2

I wish you good luck with this study!

Below I provide some detailed suggestions on how you can improve your work before resubmission:

Introduction

Authors subdivided the introduction into 3 parts, in fact. In my opinion, section 2 (aim and objective) and section 3 (why Malta) should be integrated with the current introduction (section 1). Please note that in section 2 (aim and objective) you actually explain why Malta as well. Moreover, the reference to Notre-Dame example should be reconsidered to highlight better the contribution and value of this study. Currently, you tried, but in fact the way in which you refer to Notre-Dame case states something slightly different than approached in this study…

As suggested, section 2 (aim and objective) and section 3 (why Malta) were integrated with the current introduction (section 1). The reference to Notre-Dame is reconsidered.

Finally, the introduction needs to clearly explain the purpose of the work, given the existing research gap and by this highlighting the contribution of your work. Although you try to collaborate these items it still needs substantive work.

The introduction was amended following your recommendations. The purpose of the study is clarified: to identify the risks that need to be addressed when holding fine art, determine which is perceived as being the riskiest and whether the risk perception is influenced by demographic variables.

My suggestion is to shorten the reference to Malta case in the introduction. You may use some parts of this text later on, while discussing research design.

As noted above, the reference to Malta case in the introduction is shortened.

Section 3 (which should be 4 in fact) – research questions. The research questions should be presented within the presentation of research design and method.

The research questions (RQ1 and RQ2), as well as the reasoning of these questions, are now presented in the introduction.

Section 4 – Literature review.

This section is very problematic. It is very long, there are aspects that are far from the main course of this paper. Do you really need to explain so widely what is art, fine art, definition of fraud and forgery? Please remember that your study is on insurance, so the reader expect that you will go straight to the point, rather than discussing so widely some well-established definitions (at least in the common understanding), as well as the historical contexts (eg. The text on the committed acts of theft in the past…)

Your literature review should refer only to a) valuation of art for the purposes of insurance (and please shorten the reference to social aspect!) b) the insurance of art (but please do not begin with what is insurance and the types of insurance contracts (life insurance in particular)

Instead, your literature review is missing an overview of prior works that discussed the problem of a) perception of the relevance of fine art insurance and b) the problems of underwriting in fine art insurance. In this respect, the discussion on commercial value (section 4.5.1) is incomplete – there are no reference to the literature on underwriting and the related problems while underwriting based on market value.

Following your recommendations, we have restructured the literature review, concentrating on the literature related to problems/risks faced by fine art, the objective of having Insurance and the insurances available for fine art.

Section 5 Research methods

First, this section goes beyond the presentation of research method, thus it should be titled research DESIGN and methods.

The section is renamed as suggested (Section 3. Research Design and Methods).

The fact that you plan a survey (and what is overall the scope of this survey) should be clearly communicated in the introduction as well!)

The survey (the purposely-designed questionnaire) is also mentioned in the introduction.

In lines 492-494 you stated that you refer to the risks based on the literature, but nothing was previously discussed within literature review!

To carry out the study we used a mixed-method case study approach on Malta and the Maltese people, adapting a framework suggested by Yin (2002) and Stake (1995). We identified propositions on risks related to fine art, from literature and from preliminary discussions with specialists in the field by applying a thematic analysis as suggested by Braun and Clarke (2006).

The choice of the demographic variables is based on Bezzina et al. (2012) and (2014), Savage (1993), Sund et al (2017), Weisenfeld and Ott, 2011.

Reference to Friedman test – in my opinion it would be enough to communicate that this test is compares the distribution of scores (the whole explanation of the hypotheses and the p-values is definitely not needed). Similarly, the reference to Kruskall-Wallis test. All these efforts could be exchanged by one sentence (which is communicative for researchres in this field: Due to the nature of data (survey-based) we relied on non-parametric ANOVA tests: Friedman (distribution of ranks) and Kruskal-Wallis (mean of ranks).

The references to the tests are shortened as suggested.

This section is missing the presentation of your research questions. Please note that the RQs shall correspond nicely with the further presentation of the results and the related discussion (problem by problem, not test by test)

 As recommended, we focus this study on two research questions (RQ1 and RQ2) only. The research questions are substantiated in the introduction.

Section 6. Analysis and Results

The presentation of demographics could be more communicative if you present some graphics, rather than discuss the percentage structure. From methodical point of view, this is not clear why and how you considered the age of your respondents, the level of education, insurance-background or where they work. This all should be related to prior literature (why these items are relevant for your studies).

This is even more important given the further extensive analysis of demographics and art risk perception.

The section “Analysis and Results” (now section 4) is restructured following your recommendations. 

I have a problem with how you interpreted the results of Friedman test (distributions). For me, you should first present the table 1b, then fig 1, and then the results of the test. Please note that the hypothesis you formed is far from the RQ you placed… You should highlight which RQ is answered by verification of this hypothesis.

The presentation of the study results as well as the interpretation of the Friedman test is revised.

Now I will refer to my former doubts on the lack of explanation on why you refer to the given demographic characteristics

  • Age – do the age intervals in your study are motivated by prior studies on risk perception? Was it e.g. confirmed that the younger people are more risk averse? Please also note that you actually don’t answer the core question – which “age” group has evaluated the risk as higher/lower. That is the most interesting point here. You need to demonstrate the results of post-hoc tests and ideally the mean ranks of KW test. Otherwise, this is very superficial what you did. Instead of such a wide presentation of means scores broken by the demographics, the post-hoc tests should be presented here.
  • The same refers to your further statistics. I also noted that while you were more communicative with age, the presentation of other demographics is insufficient (you only place a table… with one sentence and this sentence is actually not needed for the academics in this field, as we all know which statistics were significant)

As suggested, we have strengthened the presentation of demographics in the paper.

I am not sure if this is a right research procedure to mix EFA in the way you provided in this work. EFA is used if you don’t know what you may expect… Then you may apply other tests (e.g. KW – which you actually did later on again!). Also, why Cronbach-Alfa is presented at the very end? We typically explain this at the beginning, only to justify the ‘quality’ of the survey questions.

Following your suggestions, we focus the study of on RQ1 and RQ2, therefore EFA was taken off completely.

Honestly, I got lost on what and why you do… I cannot track the main problems (risk perception, underwriting) and the RQs you placed…

Following your recommendation, we put the focus of the study on the risks and risk perception, posing two research questions (RQ1 and RQ2).

The table 24b comes out of nowhere, I have no clue why these particular items are used as the hallmarks of the perceived risk? And how it refers to the risk perception items discussed above?

Table 24b is removed.

Finally, where is the underwriting? How your analysis refers to the problem of underwriting (the market value of fine art?)

Following your suggestions, underwriting is removed.

Section 7 – conclusions

The conclusions could be written much better. Currently, this is mainly the summary of the results (written in similar style as it was done previously).

As suggested, the conclusions are strengthened, stressing the results and its’ potential added value for the insurance industry.

Reviewer 2 Report

This paper has some originality and novelty. It is not ground-breaking and makes too many overambitious statements, e.g. it does not “generate a model” that will be useful for underwriting purposes. Being more realistic and down-to-earth would actually strengthen the paper. The main downfall is a complete lack of scholarly ability to discern what is relevant for readers and what is not. Far too much padding is thrown in. The paper should be significantly reduced in length.

Abstract: “participants’ demographic variables” is not clear. The last sentence of the abstract is cryptic: don’t say that insights are provided, tell the readers briefly what the insights are within the abstract.

Page 1, line 30: What is meant by “growing increasingly”? Is the rate of growth of the industry accelerating? How do you measure this? Sums insured? Policy values? Premium income? Claim payouts? Please substantiate with a reference.

Page 1, line 37: What does “this” refer to?

Page 1, line 43: What is “the Notre Dame”? Is it a cathedral? National heritage of what country?

Page 1, line 44: What does “these” refer to?

Page 2, lines 48-49: Clear contradiction. Notre Dame was self-insured by its owner, the state of France.

Page 2, 1st paragraph: The aims and objectives of the paper in this 1st paragraph are very poorly worded. Please explain this unambiguously.

Page 2, line 65: What is a “home policy”? This not standard insurance terminology.

Page 2, section 3: This reads like a tourist brochure. Please summarize this in one short paragraph. None of the details here are required in a cutting-edge research article.

Page 3, lines 106-7: “None of these thefts were actually insured”. How do you insure a theft?

Page 3, lines 108-112: Studies of islands? How is this at all relevant to risk and financial management? Please delete this whole paragraph and the totally irrelevant references.

Page 3: There are two sections labelled as section 3. The second section 3 is just a list with no context provided.

Page 3, line 117: What do you mean by a “factor variable”?

Pages 2-11: The contents of sections 2-4 are extremely verbose. This is just padding to make the paper appear more substantial than it actually is. Much of it is either well-known, or obvious, or covered in the literature, so you can just reference the literature. For example, much of section 4 seems to be shamelessly repeating Findlay (2014). Please summarize all of sections 2-4, keep the key ideas, and reduce to one section only, over not more than 3 pages. Readers would like to see an up-to-date scholarly review of academic research, not a description of vague websites and art history.

Page 7, line 309: Explain “a definite minute price”.

Page 7, line 328: Provide a reference for Guntram.

Pages 6-7: The footnotes add absolutely nothing to the paper.

Page 11, line 522: Explain succinctly the statistical methodology that Creative Research Systems (n.d.) uses to calculate sample sizes.

Page 12, section 5.3: This is completely unnecessary. Please delete.

Page 12, lines 550-553: This is extremely elementary. This paragraph reads like an undergraduate essay.

Page 12, first sentences of sections 5.3.1 and 5.3.2 are incomplete and poorly written sentences.

Pages 12-13, sections 5.3.1-5.3.5: Too many short subsections creates discontinuity in the flow of the text. Merge all of this in one single section 5.3 “Data analysis”.

Page 13, section 6.1: Present all of this in a Table. In the text, explain the relevance of the composition of the participant population, e.g. in terms of education.

PAge 14, Table 1a. This can be written much more concisely in the text. No Table is needed.

Page 14, Fig 1. This is very difficult to read. Split into two rows.

Page 14, section 6.3: There is no need for many microscopic subsections. Merge them all into one section.

Pages 15-18: Why do we need Tables 2, 4, 6? Tables 3, 5, 7 etc. are sufficient

Page 18, section 6.4: See my comment about section 6.3 above.

Page 19, Table 12: Please make use of something called Appendices. Table 12 is a perfect candidate for being moved to an appendix.

Pages 21-25: Why do we need Tables 14, 16, 18 etc.? Tables 15, 17, 19 etc. are sufficient.

Page 25: There are many weakly informative Tables presented. Authors need to show critical skills and be selective about the information that they present and what will be useful to readers.

Page 25, Table 44a: There are so many irrelevant Tables included in the paper, that the Table numbering system breaks down. Delete Table “44a” and write this succinctly in the text.

Page 27, Figure 2: This is completely illegible. There does not seem to be any effort from the authors to try to present their results in a way that is comprehensible.

Page 28, Table 25a: Delete this and write the contents in the text.

Page 29, Table 26: This is totally uninteresting. Please summarise your findings in one short paragraph and tell us what is novel about this.

Page 1, line 13 and Page 31, line 867: Repetition of a whole sentence: “The aim with this paper is to highlight in this article the important factors and perceived risks to consider when underwriting fine art insurance policies”. And within this repetitive sentence is the repetition of “with this paper” and “in this article”.

Page 31, line 885: Rewrite and clarify this sentence.

Page 31, line 901: “papers”

Page 31, last 2 paragraphs: Please summarise this in 2-3 lines. The authors are over-ambitious in stating that they have “generate[d] a model” and that they “provide a basis for which such a product”. You should instead discuss critically the weakness in your statistical approach and results and what work you will do in the future to address these weaknesses and to take this research forward.

Page 32, lines 925-934: What is this text? How is this relevant to the paper? Is important text missing here?

Pages 32-37: There are many references that are completely irrelevant from a risk and financial management perspective. Do you seriously need to cite tutorials on how to use SPSS? Do we really need so many references about art and art history? It is as if the authors have no critical ability to be selective about what information will be useful to readers.

Author Response

Thank you very much for your comments and recommendations, which have been much appreciated and duly noted. This has made our paper stronger. Answers and clarifications have been provided in blue below your evaluation, for ease of reading.

Hope this is to your satisfaction.

Kind regards,

The authors

This paper has some originality and novelty. It is not ground-breaking and makes too many overambitious statements, e.g. it does not “generate a model” that will be useful for underwriting purposes. Being more realistic and down-to-earth would actually strengthen the paper. The main downfall is a complete lack of scholarly ability to discern what is relevant for readers and what is not. Far too much padding is thrown in. The paper should be significantly reduced in length.

Following your recommendations, we have strengthened the paper focusing on RQ1 and RQ2. Therefore, the paper is shortened.

RQ1-      What are the risks that need to be addressed when holding art and which risks are perceived as being riskiest?

RQ2-      How do demographic variables affect people’s perception of art risks and art insurance?

Abstract: “participants’ demographic variables” is not clear. The last sentence of the abstract is cryptic: don’t say that insights are provided, tell the readers briefly what the insights are within the abstract.

As suggested, we have clarified the aim of the study. Besides, we specify that we analyse whether the risk perception is influenced by demographic variables related to the fine art owners. The insights of the study are added in the abstract.

Page 1, line 30: What is meant by “growing increasingly”? Is the rate of growth of the industry accelerating? How do you measure this? Sums insured? Policy values? Premium income? Claim payouts? Please substantiate with a reference.

As suggested, this sentence is clarified, the reference is added (Friedman et al., 2020).

Page 1, line 37: What does “this” refer to?

We agree and this sentence was deleted.

Page 1, line 43: What is “the Notre Dame”? Is it a cathedral? National heritage of what country?

We agree and the reference to Notre-Dame is reconsidered. Following your suggestion, we have added the words “famous cathedral, Notre-Dame de Paris, France”.

Page 1, line 44: What does “these” refer to?

We agree and this sentence was deleted.

Page 2, lines 48-49: Clear contradiction. Notre Dame was self-insured by its owner, the state of France.

We agree and this sentence was deleted.

Page 2, 1st paragraph: The aims and objectives of the paper in this 1st paragraph are very poorly worded. Please explain this unambiguously.

As per your suggestions, the formulation of the aim is clarified: we aim to identify the risks that need to be addressed when holding fine art and determine which is perceived as being the riskiest. Moreover, in doing this, we analyze whether the risk perception is influenced by demographic variables related to the fine art owners. Therefore, we have reduced the number of research questions to two:

RQ1-   What are the risks that need to be addressed when holding art and which risks are perceived as being riskiest?

RQ2-   How do demographic variables affect people’s perception of art risks and art insurance?

Page 2, line 65: What is a “home policy”? This not standard insurance terminology.

Following your recommendation, we have changed to standard “property-insurance policy”.

Page 2, section 3: This reads like a tourist brochure. Please summarize this in one short paragraph. None of the details here are required in a cutting-edge research article.

As suggested, we have shortened information on Malta, section 3 (why Malta) was integrated with the current introduction (section 1).

Page 3, lines 106-7: “None of these thefts were actually insured”. How do you insure a theft?

As suggested we have clarified this sentence, changing to “insuring against the thefts of fine arts”.

Page 3, lines 108-112: Studies of islands? How is this at all relevant to risk and financial management? Please delete this whole paragraph and the totally irrelevant references.

As suggested we have changed the positioning of this paragraph to flow better and to show the significance and need of such a study.

Page 3: There are two sections labelled as section 3. The second section 3 is just a list with no context provided.

We agree and this has been amalgamated to flow better from the introduction.

Page 3, line 117: What do you mean by a “factor variable”?

We agree and this was deleted since it was not relevant.

Pages 2-11: The contents of sections 2-4 are extremely verbose. This is just padding to make the paper appear more substantial than it actually is. Much of it is either well-known, or obvious, or covered in the literature, so you can just reference the literature. For example, much of section 4 seems to be shamelessly repeating Findlay (2014). Please summarize all of sections 2-4, keep the key ideas, and reduce to one section only, over not more than 3 pages. Readers would like to see an up-to-date scholarly review of academic research, not a description of vague websites and art history.

Following your suggestions, we have revised the content of sections 2-4, partially deleting or integrating the content in section 1 (introduction).

Sections were deleted and revised as per your suggestions

Page 7, line 309: Explain “a definite minute price”.

As suggested, this sentence was clarified, mentioning “definite price being paid for a possible large loss to be covered which highlights the transferring of one’s risk”.

Page 7, line 328: Provide a reference for Guntram.

We agree and this was deleted since it was not relevant as the paper was shortened.

Pages 6-7: The footnotes add absolutely nothing to the paper.

we agree and the footnotes are removed as suggested.

Page 11, line 522: Explain succinctly the statistical methodology that Creative Research Systems (n.d.) uses to calculate sample sizes.

We agree. The wording has been changed to: Using an online sample size calculator - Creative Research Systems (n.d), we determined that a sample size of 384 participants is needed at a 95% confidence level and 5% error.  In total, we collected 465 valid responses from participants who felt they could contribute to this study since they were in one way or another connected to, employed in the area of or knowledgeable about the subject of this study.

Page 12, section 5.3: This is completely unnecessary. Please delete Page 12, lines 550-553: This is extremely elementary. This paragraph reads like an undergraduate essay.Page 12, first sentences of sections 5.3.1 and 5.3.2 are incomplete and poorly written sentences. Pages 12-13, sections 5.3.1-5.3.5: Too many short subsections creates discontinuity in the flow of the text. Merge all of this in one single section 5.3 “Data analysis”.

Based on your suggestion, we have restructured this section.

Page 13, section 6.1: Present all of this in a Table. In the text, explain the relevance of the composition of the participant population, e.g. in terms of education.

We took up the suggestion and presented this as a table

PAge 14, Table 1a. This can be written much more concisely in the text. No Table is needed.

We took up the suggestion and presented this in text

Page 14, Fig 1. This is very difficult to read. Split into two rows.

Following your comment, we have removed the Figures. Figure 1 is substituted with Table 6.

Page 14, section 6.3: There is no need for many microscopic subsections. Merge them all into one section. Pages 15-18: Why do we need Tables 2, 4, 6? Tables 3, 5, 7 etc. are sufficient Page 18, section 6.4: See my comment about section 6.3 above.Page 19, Table 12: Please make use of something called Appendices. Table 12 is a perfect candidate for being moved to an appendix.Pages 21-25: Why do we need Tables 14, 16, 18 etc.? Tables 15, 17, 19 etc. are sufficient. Page 25: There are many weakly informative Tables presented. Authors need to show critical skills and be selective about the information that they present and what will be useful to readers.Page 25, Table 44a: There are so many irrelevant Tables included in the paper, that the Table numbering system breaks down. Delete Table “44a” and write this succinctly in the text.Page 27, Figure 2: This is completely illegible. There does not seem to be any effort from the authors to try to present their results in a way that is comprehensible. Page 28, Table 25a: Delete this and write the contents in the text.

We reorganised all the tables as suggested

Page 29, Table 26: This is totally uninteresting. Please summarise your findings in one short paragraph and tell us what is novel about this.

We carried out the changes suggested

Page 1, line 13 and Page 31, line 867: Repetition of a whole sentence: “The aim with this paper is to highlight in this article the important factors and perceived risks to consider when underwriting fine art insurance policies”. And within this repetitive sentence is the repetition of “with this paper” and “in this article”. Page 31, line 885: Rewrite and clarify this sentence. Page 31, line 901: “papers”. Page 31, last 2 paragraphs: Please summarise this in 2-3 lines. The authors are over-ambitious in stating that they have “generate[d] a model” and that they “provide a basis for which such a product”. You should instead discuss critically the weakness in your statistical approach and results and what work you will do in the future to address these weaknesses and to take this research forward.

We carried out the changes suggested. The text was reworded.

Page 32, lines 925-934: What is this text? How is this relevant to the paper? Is important text missing here?

We agree the test was deleted

Pages 32-37: There are many references that are completely irrelevant from a risk and financial management perspective. Do you seriously need to cite tutorials on how to use SPSS? Do we really need so many references about art and art history? It is as if the authors have no critical ability to be selective about what information will be useful to readers.

Following your recommendation, we have shortened this paragraph, removing unnecessary references.

Round 2

Reviewer 1 Report

Dear Authors,

the paper is much better - it was a good decision to keep it shorter and more concerned about the single problem.

However, there are still three important issues that need to be addressed before publication:

1) the explanation of the demographic variables. Still, there is no information on why these particular demographic variables and why in this particular intervals (e.g. age)

2) In section 4.3., starting from table 7. First, as you perform ANOVA, the statistic should be not Chi-Square, but K-W or Kruskal-Wallis. I also recommend to clearly signalise which p-values are statistically significant (only for clarity on a reader's side). This comment refers also to tables 8 - 11. Still, I cannot see the analysis of post-hoc tests that is needed to explain which particular groups of e.g. age differ significantly. For instance, if in table 7 you confirmed that there is statistically significant effect of age in legal issues arising from the ownership art works, the post-hoc tests explain which particular age groups differ in this respect? As there are only several statistically significant differences in tables 7-11, this is a matter of adding one table that reports the KW post-hoc ranks for statistically significant differences. You added the interpretation of ranks (which is good), but this must be accompanied by the post-hoc tests. Please note that the statistical software performs post-hoc tests automatically, so I see no problem to report this. 

Please have a closer look on the interpretation of data. For instance, in table 7 age/legal issues p=0,001, in the text you write 0,005???? 

Overall, the statististical soundness of this part is still low. I hope you will improve this.

3) In my opinion, in the conclusions you should clearly communicate the limitations of your study. In fact, the statistically significant differences are in minority (in some demographic variables - none). Thus, you should demonstrate your awareness on the limited evidence of this empirical analysis. 

Good luck! 

Author Response

Thank you for your comments and suggestions, which has helped to make our paper much better. We have answered below your comments in red for ease of the reader.

the paper is much better - it was a good decision to keep it shorter and more concerned about the single problem.

However, there are still three important issues that need to be addressed before publication:

1) the explanation of the demographic variables. Still, there is no information on why these particular demographic variables and why in this particular intervals (e.g. age)

Thank you for the comment – explanations have been added on page 5, lines 238-243.

2) In section 4.3., starting from table 7. First, as you perform ANOVA, the statistic should be not Chi-Square, but K-W or Kruskal-Wallis.

Thanks – this has been amended.

I also recommend to clearly signalise which p-values are statistically significant (only for clarity on a reader's side). This comment refers also to tables 8 - 11.

This has also been amended.

Still, I cannot see the analysis of post-hoc tests that is needed to explain which particular groups of e.g. age differ significantly. For instance, if in table 7 you confirmed that there is statistically significant effect of age in legal issues arising from the ownership art works, the post-hoc tests explain which particular age groups differ in this respect? As there are only several statistically significant differences in tables 7-11, this is a matter of adding one table that reports the KW post-hoc ranks for statistically significant differences. You added the interpretation of ranks (which is good), but this must be accompanied by the post-hoc tests. Please note that the statistical software performs post-hoc tests automatically, so I see no problem to report this. 

Many thanks – we have now added post-hoc analyses to Tables 7, 8 and 9.

Please have a closer look on the interpretation of data. For instance, in table 7 age/legal issues p=0,001, in the text you write 0,005???? Overall, the statististical soundness of this part is still low. I hope you will improve this.

This has been amended.

3) In my opinion, in the conclusions you should clearly communicate the limitations of your study. In fact, the statistically significant differences are in minority (in some demographic variables - none). Thus, you should demonstrate your awareness on the limited evidence of this empirical analysis. 

Thanks you

Reviewer 2 Report

Thank you for taking on board the comments, shortening the paper, and focussing it on the financial and risk management issues which are relevant to researchers.

General: People who buy insurance for fine art are those who own the fine art, or who are trustees of institutions that hold this art. It is their risk attitude that matters in terms of whether they buy insurance, at what premium, and with what deductible (policy excess). In this paper, however, you survey people who work in the art or insurance worlds. How do you explain this divergence in your study between owners and workers? For example, car mechanics may have a view about risks related to cars, but it is car owners who ultimately buy automobile insurance.

Abstract: In the abstract, you mention “demographic variables” and “individual demographics” at least 3 times, without elaborating on these. What precisely are these variables? It will benefit potential readers to know early on what are some of the key variables whose impact on risk perception you have uncovered.

Lines 16 and 102: You say that you are seeking to identify the risks that are perceived as riskiest. Are there risky risks, riskier risks, and riskiest risks? Please rephrase this.

Line 110: Are “level of education” and “field of study” demographic variables?

Lines 110 and 115: Why the repetition in close proximity about demographic variables?

Lines 134 and 135: Join the paragraphs.

Line 986: Explain succinctly the statistical methodology that Creative Research Systems (n.d.) uses to calculate sample sizes, and show that you understand the underlying statistical theory. Incidentally, the date for this citation should be the date that you accessed this web resource.

Line 1417: “while restoration”. This has to be clarified.

Line 1370: There is also no statistically significant result based on job position.

Section 4.4 is very interesting and supplies relevant information. Thank you for summarising this.

Conclusion, section 7: No survey and statistical analysis is perfect. Please show critical ability and discuss the weaknesses in your survey and statistical results. Explain also what work you will do in the future to address these weaknesses and to take this research forward.

Author Response

Thank you for your comments and suggestions, which has helped to make our paper much better. We have answered below your comments in Bold for ease of the reader.

Thank you for taking on board the comments, shortening the paper, and focussing it on the financial and risk management issues which are relevant to researchers.

General: People who buy insurance for fine art are those who own the fine art, or who are trustees of institutions that hold this art. It is their risk attitude that matters in terms of whether they buy insurance, at what premium, and with what deductible (policy excess). In this paper, however, you survey people who work in the art or insurance worlds. How do you explain this divergence in your study between owners and workers? For example, car mechanics may have a view about risks related to cars, but it is car owners who ultimately buy automobile insurance.

Many thanks for this comment. We tried to capture a wide gamut of opinions and viewpoints, particularly given the dearth of research within this specific field. We have now added clarification regarding the roles and their relationship to fine art in the paper itself – on page 4, lines 192 to 201.

Abstract: In the abstract, you mention “demographic variables” and “individual demographics” at least 3 times, without elaborating on these. What precisely are these variables? It will benefit potential readers to know early on what are some of the key variables whose impact on risk perception you have uncovered.

Thanks for this – it has been amended.

Lines 16 and 102: You say that you are seeking to identify the risks that are perceived as riskiest. Are there risky risks, riskier risks, and riskiest risks? Please rephrase this.

Thanks for this – it has been Done

Line 110: Are “level of education” and “field of study” demographic variables?

Yes, they are part of our demographics

Lines 110 and 115: Why the repetition in close proximity about demographic variables?

This has been amended

Lines 134 and 135: Join the paragraphs.

Thanks for this – it has been Done

Line 986: Explain succinctly the statistical methodology that Creative Research Systems (n.d.) uses to calculate sample sizes, and show that you understand the underlying statistical theory. Incidentally, the date for this citation should be the date that you accessed this web resource.

Thanks for this – it has been Done

Line 1417: “while restoration”. This has to be clarified.

Thanks for this – it has been Done

Line 1370: There is also no statistically significant result based on job position.

Thank you – this has been added

Section 4.4 is very interesting and supplies relevant information. Thank you for summarising this.

Conclusion, section 7: No survey and statistical analysis is perfect. Please show critical ability and discuss the weaknesses in your survey and statistical results. Explain also what work you will do in the future to address these weaknesses and to take this research forward.

Thank you for this – we have added a paragraph on page 20-21, lines 460-469.